# Staggered Rollout Designs Enable Causal Inference Under Interference Without Network Knowledge

**Mayleen Cortez-Rodriguez**
Center for Applied Math
Cornell University
Ithaca, NY 14850
mec383@cornell.edu

**Matthew Eichhorn**
Center for Applied Math
Cornell University
Ithaca, NY 14850
meichhorn@cornell.edu

**Christina Lee Yu**
Operations Research and Information Engineering
Cornell University
Ithaca, NY 14850
cleeyu@cornell.edu

## Abstract

Randomized experiments are widely used to estimate causal effects across many domains. However, classical causal inference approaches rely on independence assumptions that are violated by *network interference*, when the treatment of one individual influences the outcomes of others. All existing approaches require at least approximate knowledge of the network, which may be unavailable or costly to collect. We consider the task of estimating the total treatment effect (TTE), the average difference between the outcomes when the whole population is treated versus when the whole population is untreated. By leveraging a *staggered rollout* design, in which treatment is incrementally given to random subsets of individuals, we derive unbiased estimators for TTE that *do not rely on any prior structural knowledge of the network*, as long as the network interference effects are constrained to low-degree interactions among neighbors of an individual. We derive bounds on the variance of the estimators, and we show in experiments that our estimator performs well against baselines on simulated data. Central to our theoretical contribution is a connection between staggered rollout observations and polynomial extrapolation.

## 1   Introduction

A cornerstone of much of the classic causal inference literature is the stable unit treatment value assumption (SUTVA), which posits that an individual's potential outcome is a function only of their assigned treatment; there are no spillover effects due to the treatment of others. Such an assumption fails to account for the ways in which individuals interact in many real-world experimental settings. For instance, new features rolled out on social networking sites such as LinkedIn may alter these users' behaviors, which in turn affect how their connections (who do not have access to the feature) interact with the platform. Individuals receiving a vaccine against an infectious disease may reduce the transmission probability of the disease to others they interact with. Implementing a different pricing policy for a subset of individuals in an online marketplace such as Airbnb or a platform such as Uber could impact the experience of other users, as they compete for the same resources or same customers. Public health measures instituted in one city can limit travel to nearby communities, indirectly affecting their health outcomes or transit related outcomes. These examples illustrate how

36th Conference on Neural Information Processing Systems (NeurIPS 2022).

network interference may arise naturally from the connectedness of our society. Unfortunately, the standard causal inference techniques which do not account for network interference may result in arbitrarily biased estimates.

As these issues come into greater focus, there is a growing research area in developing new tools for causal inference under *network interference*, when the outcome of an individual can be affected by the treatment of another. Many approaches either propose complex graph-based cluster randomized designs, or require strong parametric assumptions on the network interference effects. A limitation is that all these approaches require at least partial knowledge of the underlying network in order to implement the randomization or to compute the estimator. While structural knowledge is available to online social networks, other applications such as public health must reason about an unknown or potentially transient network. The additional effort required to collect or model network structure is both difficult and costly.

In this work, we explore the value of additional measurements which arise from a staggered rollout randomized design, in which the treatments are administered over a span of a few timepoints. For example, the experimentation team at LinkedIn may roll out an experiment over 5 days, increasing the fraction of treated individuals according to a schedule of 1%, 2%, 5%, 10%, 20%, where it continuously collects data and measurements before and during each day of the experiment. Not only is this type of experiment easy to implement in such applications, it is often desirable to implement treatments according to such a staggered rollout design as it allows the system to first ensure safety of the proposed treatment on a smaller test group before implementing it on larger groups. This type of experimental design is also common for trials involving healthcare and medicine due to the requirement of certain safety considerations before testing for efficacy. A key contribution in our work is that we show the additional measurements from a *staggered rollout* design enable graph agnostic causal inference, lifting all requirements on knowledge of the network.

We focus on estimating the *total treatment effect* (TTE), informally defined as the difference in average outcomes across the population between two scenarios: when all individuals are treated and when no individuals are treated. It has also been referred to as the *global average treatment effect* (GATE). The TTE is particularly pertinent to applications where the decision maker must choose between entirely adopting the new treatment or remaining with the status quo. For example, LinkedIn would like to choose a single news feed recommendation algorithm, and Airbnb and Uber would like to choose a single dynamic pricing algorithm. We assume *neighborhood interference*, where each individual is only affected by the treatments of its direct neighbors; this is only mildly restrictive as the neighborhood can be defined with respect to an unknown network, which is neither used for the estimator nor the randomized design.

**Related Work.**  In addressing the challenges that arise from network interference, a key tension arises between the model assumptions and the simplicity and efficiency of the proposed estimator. Previously proposed model assumptions can be generally classified into assumptions on exposure functions [1, 2, 12, 14, 24], interference neighborhoods [3, 5, 18, 22], parametric structure [4, 6, 8, 9, 20], or a combination of these. Each of these assumptions lead to different solution concepts. All of these approaches rely on knowledge of the network mediating the interference effects.

One class of approaches relies on assumptions about the network structure. They assume *partial interference*, meaning that the population can be partitioned into disjoint groups, such that all network interference effects can only occur within but not across the pre-specified groups [2, 5, 10, 13, 16, 17, 19, 23]. This assumption is motivated by scenarios where the network is naturally strongly clustered. A natural solution is to randomize treatments over the groups jointly, such that each group is assigned to be either fully treated or fully control. A drawback of this approach is that many networks are well-connected such that there is no clear clustering of the network which does not cut a significant fraction of the edges. The bias of standard estimators will scale with the number of edges that cross between groups, leading to proposed cluster randomized designs that randomize over clusters that are constructed to minimize the number of edges between clusters [8, 9]. Constructing good clusters itself can be computationally intensive. Additionally some applications may prohibit such nonuniform treatment assignment probabilities due to fairness considerations. Under *neighborhood interference* assumptions, [21, 22] analyze the Horvitz-Thompson estimator alongside a cluster randomized design, which involves both clustering the graph and computing probabilities of entire neighborhoods being assigned to treatment or control over the distribution of clusterings, which is computationally intensive. When one is willing to impose a distributional model on the network itself,

[11] provides central limit theorem convergence results for a related but weaker estimand measuring the change in outcomes under small perturbations of the fraction of treated individuals.

An alternate approach is to impose structure on the form of the network interference effects. The most common assumption is that the network effects are linear with respect to a specified statistic of the local neighborhood [4, 6, 7, 9, 15, 20]. The assumptions reduce the number of unknown parameters in the model to a fixed dimension that does not grow with the population size, reducing the inference task to linear regression. As a result, the natural solution is to use a least squares estimate, shifting the focus to constructing randomized designs that minimize the variance of the estimate. A limitation of this approach is that it requires the correct choice of the the statistic governing the linearity, and it requires precise knowledge of the network structure to compute these neighborhood statistics. Furthermore, it assumes knowledge of the relevant covariate types that differentiate individual responses, or otherwise assumes homogeneity in the network effects.

The most similar work to our paper is the solution proposed in [25], which provides an estimator for the TTE under a heterogeneous linear interference model [8], also referred to as the joint assumptions of additivity of main effects and interference effects in [18]. Their estimator does not require knowledge of the network, but requires measurements over two time steps. Our work generalizes their results beyond linear to polynomial models, and we show that the staggered rollout experimental design enables graph agnostic causal inference. The extension from linear to polynomial models introduces the possibility of non-trivial interactions within the neighborhood set, adding complexity to the model that necessitates a new analysis and algorithm.

**Contributions.** We show that under a staggered rollout experimental design, the task of estimating the total treatment effect reduces to polynomial extrapolation, where the degree of the polynomial is governed by the cardinality of interactions in the neighborhood interference model, bounded above by the degree of the graph. Our approach is the first in the literature to propose an estimator and randomized design that does not require any knowledge of the network structure, and yet is unbiased and consistent. We provide variance bounds on the estimator, showing that the variance only grows polynomially in the degree as opposed to the exponential growth that is exhibited in the Horvitz-Thompson estimator under simple Bernoulli randomized designs. We provide experiments that also illustrate that naively using regression models without allowing for heterogeneity could lead to significant bias, whereas our estimator is unbiased with significantly lower variance than the bias incurred due to a misspecified model. We are also the first to study the value of a staggered rollout experimental design in the presence of network interference, and we believe the overall framework could extend beyond polynomial models to other function classes, opening a new approach for handling network interference while allowing for flexible heterogeneity in the network effects.

## 2 Setup

Consider a population of $n$ individuals, and assume that the network interference can be represented via an unknown directed graph with edge set $E \subset [n] \times [n]$. An edge $(j, i) \in E$ represents that individual $i$ is affected by the treatment assignment of individual $j$; as such, self-loops are expected. The in-neighborhood of individual $i$ is denoted by $\mathcal{N}_i = \{j \in [n] : (j, i) \in E\}$, and we let $d_{\text{in}}$ denote the maximum in-degree, $d_{\text{out}}$ the maximum out-degree, and $d = \max\{d_{\text{in}}, d_{\text{out}}\}$. We posit that the outcome of individual $i$ as a function of the entire population's exposure to treatment can be expressed by the potential outcomes function $Y_i : \{0, 1\}^n \to \mathbb{R}$.

Our task is to estimate the total treatment effect (TTE), which represents the difference in average outcomes when the entire population is fully under treatment as opposed to fully under control, denoted as

$$\text{TTE} := \tfrac{1}{n} \sum_{i=1}^n \big( Y_i(\mathbf{1}) - Y_i(\mathbf{0}) \big). \tag{2.1}$$

We use $\mathbf{z} \in \{0, 1\}^n$ to denote the treatment assignment vector, where $z_i = 1$ if individual $i$ is assigned to treatment, and $z_i = 0$ if $i$ is assigned to control. By definition of $E$, it follows that the potential outcomes functions satisfy neighborhood interference with respect to the graph defined by $E$.

**Assumption 1** (Neighborhood Interference). *$Y_i(\mathbf{z})$ only depends on the treatment of individuals in $\mathcal{N}_i$ (including $i$). Equivalently, $Y_i(\mathbf{z}) = Y_i(\mathbf{z}')$ for any $\mathbf{z}$ and $\mathbf{z}'$ such that $z_j = z'_j$ for all $j \in \mathcal{N}_i$.*

Additionally, as the treatment variables $z_i$ are binary, any potential outcomes function satisfying neighborhood interference can be written as a polynomial in the neighborhood treatment variables:

$$Y_i(\mathbf{z}) = \sum_{\mathcal{S} \subseteq \mathcal{N}_i} a_{\mathcal{S}} \prod_{j \in \mathcal{S}} z_j \prod_{j' \in \mathcal{N}_i \setminus \mathcal{S}} (1 - z_{j'}),$$

for some coefficients $\{a_{\mathcal{S}}\}$. We use the degree of the polynomial to quantify the complexity of the model. In full generality, any model satisfying the neighborhood interference assumption will have polynomial degree bounded by $\max_i |\mathcal{N}_i|$, the maximum in-degree of the graph. In this work we consider the scenario where the polynomial degree may be significantly smaller than $\max_i |\mathcal{N}_i|$.

**Assumption 2** (Low Polynomial Degree). *The potential outcomes model has polynomial degree at most $\beta$, i.e. there exist coefficients $\{c_{i,\mathcal{S}}\}_{i \in [n], \mathcal{S} \subseteq [n]}$ such that for all $i$ and $\mathbf{z}$,*

$$Y_i(\mathbf{z}) = \sum_{\mathcal{S} \subseteq \mathcal{N}_i, |\mathcal{S}| \leq \beta} c_{i,\mathcal{S}} \cdot \mathbb{I}(\mathcal{S} \text{ treated}) = \sum_{\mathcal{S} \subseteq \mathcal{N}_i, |\mathcal{S}| \leq \beta} c_{i,\mathcal{S}} \prod_{j \in \mathcal{S}} z_j. \qquad (2.2)$$

The low-degree polynomial structure is perhaps better conceptualized as a constraint on the order of interactions among neighbors of an individual, as the potential outcomes function is polynomial with respect to the *binary* treatment vector. For general $\beta$, this assumption is not restrictive at all; rather, a restriction is imposed by assuming a specific value for $\beta$, or more generally assuming that $\beta$ is much smaller than the graph degree. We interpret the parameter $c_{i,\mathcal{S}}$ as the effect that treating all individuals in $\mathcal{S}$ has on the outcome of individual $i$. The coefficient $c_{i,\emptyset}$ represents individual $i$'s outcome when everyone is assigned to control (i.e. their *baseline outcome*); this is unaffected by the treatment assignment. In the case of a singleton set $\mathcal{S} = \{j\}$, we use the shorthand notation $c_{ij} := c_{i,\{j\}}$. It follows that the total treatment effect is the sum of all $c_{i,\mathcal{S}}$ for nonempty subsets $\mathcal{S}$, i.e. TTE $= \frac{1}{n} \sum_{i=1}^n \sum_{\substack{\mathcal{S} \subseteq \mathcal{N}_i \\ 1 \leq |\mathcal{S}| \leq \beta}} c_{i,\mathcal{S}}$.

The number of unknown parameters in this model are $\sum_{i \in [n]} \sum_{k=0}^{\beta} \binom{|\mathcal{N}_i|}{k}$, which scales as $nd^{\beta}$. When $\beta = 1$, the network effects resulting from treated neighbors is additive, and is also equivalent to the heterogeneous linear outcomes model in [25]. This low degree assumption will not generally admit threshold models or saturation models, both of which would require the degree of $Y_i$ to be $|\mathcal{N}_i|$.

An example in which the polynomial degree may be smaller than the neighborhood size would be a setting in which an individual's neighborhood can be further partitioned into smaller subcommunities: colleagues, university friends, high school friends, family, etc. Each subcommunity could have an additive affect on the individuals' outcome, but there may be nontrivial interactions among the treatments of individuals in the subcommunities. The polynomial degree would be bounded by the size of the largest subcommunity, which could be significantly smaller than the full neighborhood. As another example, suppose that a social platform is testing a "hangout room" feature that provides groups of up to 5 people a new environment to engage on the platform. One could posit a model with $\beta = 5$, as the change in any individual's usage on the platform can be attributed to experience with the new feature, which takes place in groups of up to 5 users.

We let $Y_{\max}$ denote an upper bound on the absolute treatment effects for each individual, i.e.

$$Y_{\max} := \max_{i \in [n]} \sum_{\mathcal{S} \subseteq \mathcal{N}_i, |\mathcal{S}| \leq \beta} |c_{i,\mathcal{S}}|.$$

It follows that the magnitude of the outcomes $Y_i(\mathbf{z})$ are bounded by $Y_{\max}$ for any treatment vector $\mathbf{z}$. We let $L_j$ denote the absolute effect or influence that individual $j$ has on the population outcomes,

$$L_j := \sum_{i:j \in \mathcal{N}_i} \sum_{\mathcal{S} \subseteq \mathcal{N}_i, |\mathcal{S}| \leq \beta, j \in \mathcal{S}} |c_{i,\mathcal{S}}|.$$

Our boundedness assumption and the finiteness of our network imply the boundedness of the $L_j$. We denote the upper bound on the absolute effect or influence of any individual by $L_{\max} := \max_j \{L_j\}$.

**Randomized Experiment Design.** As it may be costly and/or detrimental to expose the entire population to treatment, we wish to estimate the total treatment effect after treating only a small random subset of individuals. In particular we assume that there may be an experimental budget that limits the proportion of individuals who may be treated. We will focus on two standard randomized designs. In Bernoulli design, a treatment vector $\mathbf{z}$ is obtained by independently sampling each

coordinate from a Bernoulli($p$) distribution, so that the probability that a subset of individuals $\mathcal{S}$ are all treated is $p^{|\mathcal{S}|}$. We assume that $p > \frac{1}{n}$ so that at least one individual is treated in expectation. In completely randomized design, a treatment vector $\mathbf{z}$ is obtained by uniformly sampling a subset of $k$ individuals to treat for some fixed $k$. Here, the probability that a subset of individuals $\mathcal{S}$ are all treated is

$$\prod_{i=0}^{|\mathcal{S}|-1} \frac{k-i}{n-i} =: \left[\frac{k}{n}\right]^{|\mathcal{S}|}. \tag{2.3}$$

Throughout the paper, we utilize a *staggered rollout* experimental design. Treatment is assigned to individuals in $T$ stages throughout the experiment. Overall, the individuals' outcomes are measured $T+1$ times: a baseline measurement before treatment, as well as a measurement after each treatment round. We'll use $\mathbf{z}^t$ to denote the vector of treatment assignment in round $t$, and assume that each entry $z_i^t$ is monotone increasing with $t$ (individuals cannot be un-treated). This monotonicity requirement introduces significant correlation between the treatment vectors. Monotonicity is a constraint in many real-world scenarios where the experimental designer only has the option to introduce treatment to new individuals. For example, treatments in medication trials can have life-altering, irreversible effects, and the exposure of individuals to an advertising campaign cannot be "taken back." In other domains, such as the rollout of new interfaces on social media platforms, treatments are temporary or reversible and it may make sense to remove the monotonicity requirement.

Another assumption we make is that observations of outcomes are perturbed by iid Gaussian noise.

**Assumption 3.** *We observe* $Y_{i,t}^{\mathrm{obs}} = Y_i(\mathbf{z}^t) + \varepsilon_{i,t}$ *for* $\varepsilon_{i,t} \overset{iid}{\sim} N(0, \sigma^2)$.

# 3 Graph Agnostic Estimators under Staggered Rollout Design

To motivate the design of our estimators, we begin with a high-level view of estimating the total treatment effect. We limit our attention to static networks. When we have no information about the underlying causal network, we do not know how much of each individual's neighborhood is treated, so have no systematic way to predict what their potential outcome would be if the entire population were treated. However, we can aggregate the average of the individuals' outcomes to obtain a meaningful statistic. In the following discussion we omit observation noise to make the intuition for our estimator clear. Consider the expected population average outcomes where the expectation is taken over the distribution of treatment vectors $\mathbf{z}$ sampled from a parameterized class of distributions $\mathcal{D}_x$, where $\mathcal{D}_0$ refers to the distribution that deterministically assigns all individuals to control, and $\mathcal{D}_1$ refers to the distribution that deterministically assigns all individuals to treatment. Consider the underlying expected outcome function $F_{\mathcal{D}} : [0,1] \to \mathbb{R}$ given by

$$F_{\mathcal{D}}(x) = \mathbb{E}\left[\frac{1}{n}\sum_{i=1}^n Y_i(\mathbf{z})\right]$$

where the expectation is taken over the distribution of treatment vectors $\mathbf{z} \sim \mathcal{D}_x$. By construction, the TTE is exactly $F_{\mathcal{D}}(1) - F_{\mathcal{D}}(0)$.

If we can implement a staggered rollout design where at stage $t$ of the experiment, the marginal distribution of the treatment vector is $\mathcal{D}_{x_t}$, the observed average outcomes collected in the experiment at stage $t$ would give noisy estimates of $F_{\mathcal{D}}(x_t)$. Under this framing, our goal is to use these measurements to extrapolate the value of $F_{\mathcal{D}}(1)$. This provides a general framework for utilizing staggered rollout design to simplify estimation of the total treatment effect.

The simplest class of distributions we can consider is the Bernoulli($p$) randomized design, in which each individual is independently assigned to treatment or control with probability $p$. For a degree-$\beta$ polynomial potential outcomes model, the expected outcome function under this design is polynomial in the treatment probability $p$:

$$F_B(p) = \mathbb{E}\left[\frac{1}{n}\sum_{i=1}^n Y_i(\mathbf{z})\right] = \frac{1}{n}\sum_{i=1}^n \sum_{\substack{\mathcal{S} \subseteq \mathcal{N}_i \\ |\mathcal{S}| \leq \beta}} c_{i,\mathcal{S}} \cdot \mathbb{E}\left[\prod_{j \in \mathcal{S}} z_j\right] = \frac{1}{n}\sum_{i=1}^n \sum_{\substack{\mathcal{S} \subseteq \mathcal{N}_i \\ |\mathcal{S}| \leq \beta}} c_{i,\mathcal{S}} \cdot p^{|\mathcal{S}|}.$$

To implement a staggered rollout Bernoulli design with treatment probabilities $p_1 < p_2 < \ldots < p_T$ we independently sample $u_i \sim U[0,1]$ for each individual $i$. Then, for each $t \in [T]$, we define treatment vector $\mathbf{z}^t$ with $\mathbf{z}_i^t = \mathbb{I}(u_i \leq p_t)$. This both ensures that the marginal distribution of the

treatment vector at stage $t$ is equivalent to the Bernoulli($p_t$) randomized design, and that the treatment assignments are monotone over the rounds.

Alternatively, we can consider a completely randomized design (CRD) in which we fix a number of treated individuals $k$, and sample a subset of $k$ individuals uniformly at random among all size $k$ subsets in the population. For a degree-$\beta$ polynomial potential outcomes model, the expected outcome function under this design is polynomial in the treated fraction $k/n$:

$$F_C(\tfrac{k}{n}) = \mathbb{E}\Big[\frac{1}{n}\sum_{i=1}^n Y_i(\mathbf{z})\Big] = \frac{1}{n}\sum_{i=1}^n \sum_{\substack{\mathcal{S}\subseteq\mathcal{N}_i \\ |\mathcal{S}|\leq\beta}} c_{i,\mathcal{S}}\cdot\mathbb{E}\Big[\prod_{j\in\mathcal{S}} z_j\Big] = \frac{1}{n}\sum_{i=1}^n \sum_{\substack{\mathcal{S}\subseteq\mathcal{N}_i \\ |\mathcal{S}|\leq\beta}} c_{i,\mathcal{S}}\cdot\Big[\frac{k}{n}\Big]^{|\mathcal{S}|}.$$

To implement a complete staggered rollout design, we sample a treatment vector from CRD($k_1$) at stage 1, and at stage $t>1$, we sample a treatment vector from CRD($k_t - k_{t-1}$) out of the remaining untreated individuals. The marginal distribution of the treatment vector at state $t$ will be equivalent to the completely randomized design with parameter $k_t$.

To construct our estimators, we will make use of the Lagrange interpolation formula.

**Definition 1** (Lagrange Interpolation). *Given a dataset $\big\{(x_t, y_t)\big\}_{t=0}^T$ with distinct x-coordinates, the unique polynomial $F$ of degree at most $T$ with $F(x_t) = y_t$ for each $t$ is given by*

$$F(x) = \sum_{t=0}^T \ell_{t,\mathbf{x}}(x)\cdot y_t, \qquad \ell_{t,\mathbf{x}}(x) = \prod_{\substack{s=0 \\ s\neq t}}^T \frac{x - x_s}{x_t - x_s}.$$

To estimate TTE, we require estimates of $F_B(x)$ or $F_C(x)$ at $x \in [0,1]$. As both $F_B$ and $F_C$ have degree at most $\beta$, they can be recovered from $\beta+1$ observations, requiring $T = \beta$ rounds of treatment. Given treatment targets $\mathbf{x} = (x_0, x_1, \ldots x_T)$ with realized treatment schedule $\{\mathbf{z}_t \sim \mathcal{D}_{x_t}\}$, we can utilize Lagrange interpolation to derive the following polynomial interpolation (PI) estimator:

$$\widehat{\mathrm{TTE}}_{\mathrm{PI}}(\mathbf{x}) := \begin{cases} \sum_{t=0}^T \big(\ell_{t,\mathbf{x}}(1) - \ell_{t,\mathbf{x}}(0)\big)\big(\frac{1}{n}\sum_{i=1}^n Y_{i,t}^{\mathrm{obs}}\big) & x_0 < x_1 < \ldots < x_T, \\ 0 & x_t = x_{t-1} \text{ for some } t \in [T]. \end{cases} \tag{3.1}$$

The separation into cases ensures that the Lagrange coefficients are well-defined. We assume that the degree $\beta$ is known such that the experimenter can select $T = \beta$. We also assume that $\mathbf{x}$ is monotone, and define $\Delta_{\mathbf{x}} = \min_{t=1..m}\big\{x_t - x_{t-1}\big\}$. We can apply this estimator in both the Bernoulli and completely randomized design settings.

## 3.1 Theoretical Results and Discussion

For a potential outcomes model with degree $\beta$, we let the notation BRD($\mathbf{p}$) refer to a staggered rollout Bernoulli design with distinct treatment probabilities $\mathbf{p} = (p_0, p_1, \ldots, p_\beta)$. We let CRD($\mathbf{k}$) refer to a staggered rollout completely randomized design with distinct treatment counts $\mathbf{k} = (k_0, k_1, \ldots, k_\beta)$.

**Theorem 2.** *Consider a potential outcomes model with degree $\beta$. Under a BRD($\mathbf{p}$) with $p_0 = 0$, the estimator $\widehat{\mathrm{TTE}}_{\mathrm{PI}}(\mathbf{p})$ is unbiased with variance*

$$O\Big(\frac{d^2\beta^2}{n}Y_{\max}^2\Delta_{\mathbf{p}}^{-2\beta} + \frac{\sigma^2\beta}{n}\Delta_{\mathbf{p}}^{-2\beta}\Big).$$

**Theorem 3.** *Consider a potential outcomes model with degree $\beta$. Under a CRD($\mathbf{k}$) with $k_0 = 0$, the TTE estimator $\widehat{\mathrm{TTE}}_{\mathrm{PI}}(\mathbf{k}/n)$ is unbiased with variance*

$$O\Big(\beta^2\, Y_{\max}^2\Big(\frac{d^2}{n} + \frac{\beta^2}{k_1}\Big)\cdot\Big(\frac{n}{\Delta_{\mathbf{k}}}\Big)^{2\beta} + \frac{\sigma^2\beta}{n}\Big(\frac{n}{\Delta_{\mathbf{k}}}\Big)^{2\beta}\Big).$$

Proofs of both of these theorems are given in Appendix **??**. Notably, families of networks with $d = o(\log n)$ have variance asymptotically approaching 0. As such, our results can generally handle sparse networks. A key technical piece of the analysis is handling the strong correlation in the observations across measurements due to the monotonicity enforced by the staggered rollout design. In the case of a linear potential outcomes model, we can strengthen both of these variance bounds, which match the results from [25], differing only in an additive term coming from observation noise.

**Corollary 4.** *For a linear potential outcomes model:*

- *The estimator $\widehat{\mathrm{TTE}}_{\mathrm{PI}}(\mathbf{p})$ under* $\mathrm{BRD}(0,p)$ *has variance at most* $\frac{1-p}{np} \cdot L_{\max}^2 + \frac{2\sigma^2}{np^2}$.

- *The estimator $\widehat{\mathrm{TTE}}_{\mathrm{PI}}(\mathbf{k}/n)$ under* $\mathrm{CRD}(0,k)$ *has variance at most* $\frac{n-k}{(n-1)k} \cdot L_{\max}^2 + \frac{2\sigma^2 n}{k^2}$.

Observe that the Bernoulli estimator does not incorporate any information about the realized treatments. Notably, it does not account for the number of treated individuals. While this binomial random variable concentrates around its mean (especially for large values of $n$), it fails to account for significant deviations from this mean. Since this information is available at the time of estimation, it can be incorporated into an estimator. We let $\hat{\mathbf{k}} = (\hat{k}_0 = 0, \hat{k}_1, \ldots, \hat{k}_\beta)$ be the realized number of treated individuals at each time step, and consider the estimator $\widehat{\mathrm{TTE}}_{\mathrm{PI}}(\hat{\mathbf{k}}/n)$.

**Theorem 5.** *Consider a potential outcomes model with degree $\beta$. Under a staggered rollout Bernoulli design with treatment probabilities $\mathbf{p} = (p_0 = 0, \ldots, p_\beta)$, $\widehat{\mathrm{TTE}}_{\mathrm{PI}}(\hat{\mathbf{k}}/n)$ has bias decaying exponentially in $n$ and variance $O\left(\beta^2\, Y_{\max}^2\left(\frac{d^2}{n} + \frac{\beta^2}{p_1 n}\right) \cdot \Delta_{\mathbf{p}}^{-2\beta}\ +\ \frac{\beta\sigma^2}{n}\Delta_{\mathbf{p}}^{-2\beta}\right)$.*

A proof is given in Appendix **??**. For large $n$, the performance of these three estimators will converge to each other. While our theoretical variance bound in Theorem 5 does not show improvement upon that from Theorem 2, our experimental results illustrate empirical improvements of this estimator.

**Discussion.** Our results illustrate a natural relationship between the complexity of the model (i.e. its degree $\beta$) and the complexity of the randomized design and corresponding estimator; we require $T \geq \beta$, i.e. $\beta + 1$ outcome measurements, in order to construct an unbiased estimator. Intuitively, each of these measurements allows us to quantify one "degree" of the network effects. Given an overall treatment budget $p = p_T$ with a uniform treatment schedule where $p_t = tp/T$, the difference between treatment fractions is $\Delta_{\mathbf{p}} = p/T$. As a result, for our setting in which $T = \beta$, the variance scales as $(\beta/p)^{2\beta}$, where $\beta$ is always bounded above by the size of the neighborhood, i.e. graph degree. In comparison, under a fully general neighborhood model, the Horvitz-Thompson estimator has a variance that scales as $O(1/np^d)$, where $d$ denotes the size of the largest neighborhood.

A practical question, critical to real-world experimental settings, is how one should determine the degree $\beta$ if it is not known in advance. Even if we have many measurements, it may not always be wise to increase the degree of the interpolant, as this increases the magnitude of its slope outside of the interpolating region $[0,p]$. When the expected number of treated individuals $np$ is small relative to the population size $n$ (so $p \ll 1$), the value of the interpolant at 1 will be highly sensitive to any deviation of the later measurements from their expectation. On the other hand, choosing to fit a low degree polynomial may lead to bias if the underlying network effects exhibit higher order interactions. The study of the sensitivity of polynomial interpolation estimators under model misspecification in our randomized experiment setting is a captivating direction for future work. In heuristic settings we recommend choosing a conservative $\beta$ erring on lower values, as is also common practice when using polynomial regression in supervised learning settings.

The low polynomial structure is primarily used to show that the expected total outcomes function $F_{\mathcal{D}}$ is a low degree polynomial of the treatment fraction. As $F_{\mathcal{D}}$ represents an expectation taken over the population, where treatments are assigned uniformly at random, it is plausible that this function varies in a smooth and simple way when the treatment level is changed. While the polynomial class is not the only hierarchy of function classes to capture complexity, it is a fairly natural one. However, continued study of this overall approach of interpolation for staggered rollout designs for other function classes beyond polynomial would also be incredibly interesting and relevant.

In this work, we also limit our attention to static network effects, but extensions to incorporate time-varying network effects or even time-varying network structures is an interesting direction for future work. When the total treatment budget $p_T$ is small, such that a significant number of individuals are observed under the baseline outcomes across all stages of the experiment, then we could handle time-fixed effects via a simple modification of our estimator by using these baseline individuals to estimate the time-fixed effects and subtracting them from the current estimator.

# 4 Experiments

We provide simulations on synthetic data to illustrate the performance of our estimators relative to existing estimators. For a population of $n$ individuals, we generate random directed networks of $n$ nodes using a configuration model with in-degrees distributed as a power law with exponent 2.5, and out-degrees evenly shared among individuals. For degree $\beta$, we construct the following potential outcomes model:

$$Y_i(\mathbf{z}) = c_{i,\emptyset} + \sum_{j \in \mathcal{N}_i} \tilde{c}_{ij} z_j + \sum_{\ell=2}^{\beta} \left( \frac{\sum_{j \in \mathcal{N}_i} \tilde{c}_{ij} z_j}{\sum_{j \in \mathcal{N}_i} \tilde{c}_{ij}} \right)^{\ell}, \tag{4.1}$$

where $c_{i,\emptyset} \sim U[0,1]$, $\tilde{c}_{ii} \sim U[0,1]$, and for $i \neq j$, $\tilde{c}_{ij} = v_j |\mathcal{N}_i| / \sum_{k:(k,j) \in E} |\mathcal{N}_k|$ for $v_j \sim U[0,r]$, where $r$ denotes a hyperparameter that governs the relative magnitude of the network effects relative to the direct effects. Essentially $v_j$ represents the magnitude of individual $j$'s influence, which is then shared among its out-neighbors proportional to their in-degrees. For simplicity we assume no observation noise in the experiments, i.e. $\sigma = 0$.

**Other Algorithms.** We benchmark our proposed estimators against least squares regression and difference-in-mean estimators. As these estimators don't utilized the staggered rollout design, we evaluate them on the measurements taken at the last stage, $T$, of the experiment. We will use $\mathbf{z}$ to denote the treatment vector at time $T$ (suppressing the superscript). As a network sampled from a configuration model does not exhibit clustering, the solutions that propose cluster randomized designs perform poorly, and thus we omit them from the experiments.

The standard difference in means estimator is the difference between the average outcome of individuals assigned to treatment and the average outcome of individuals assigned to control, given by

$$\widehat{\text{TTE}}_{\text{DM}} = \frac{\sum_{i \in [n]} z_i Y_i(\mathbf{z})}{\sum_{i \in [n]} z_i} - \frac{\sum_{i \in [n]} (1 - z_i) Y_i(\mathbf{z})}{\sum_{i \in [n]} (1 - z_i)}. \tag{4.2}$$

This estimator is biased under the presence of network interference. Note that $\widehat{\text{TTE}}_{\text{DM}}$ does not take into account any information about each individual's neighborhood.

A modification of the difference in means estimator incorporates knowledge of the number of treated neighbors of each individual. Let $U_i$ denote the number of individuals in $\mathcal{N}_i \setminus \{i\}$ assigned to treatment, and let $\tilde{U}_i$ denote the number of neighbors individuals in $\mathcal{N}_i \setminus \{i\}$ assigned to control. This estimator is given by

$$\widehat{\text{TTE}}_{\text{DM}(\lambda)} = \frac{\sum_{i \in [n]} z_i \mathbb{I}(U_i \geq \lambda) Y_i(\mathbf{z})}{\sum_{i \in [n]} z_i \mathbb{I}(U_i \geq \lambda)} - \frac{\sum_{i \in [n]} (1 - z_i) \mathbb{I}(\tilde{U}_i \geq \lambda) Y_i(\mathbf{z})}{\sum_{i \in [n]} (1 - z_i) \mathbb{I}(\tilde{U}_i \geq \lambda)}, \tag{4.3}$$

for some user-defined tolerance $\lambda \in [0, 1]$. This estimator only counts an individual's outcome if at least $\lambda$ of the individual's neighborhood is assigned to the same treatment as the individual itself. In our experiments, we set $\lambda = 0.75$.

Finally we compare against least squares regression models of degree $\beta$, which posit that the potential outcomes model can be described as

$$Y_i(\mathbf{z}) = g(z_i, \bar{z}_i) = \left( \rho + \sum_{k=1}^{\beta} \gamma_k X_i^k \right) + z_i \left( \tilde{\rho} + \sum_{k=1}^{\beta-1} \tilde{\gamma}_k X_i^k \right), \tag{4.4}$$

for some covariate $X_i$. In the two variations we consider, we set $X_i$ equal to either the number of treated neighbors or the proportion of treated neighbors, where we do not include $i$ itself. The two sets of coefficients $(\rho, \gamma_1, \ldots \gamma_\beta)$ and $(\rho, \tilde{\gamma}_1, \ldots \tilde{\gamma}_\beta)$ allow for the model to be different when $i$ is treated vs not treated, and the second summation only goes until $\beta - 1$ since we want to only allow degree $\beta$ interactions. The total number of coefficients in the model is $2\beta + 1$. Least squares regression finds the set of coefficients that minimizes the least squares predictive error on the dataset, which consists of $\{z_i, X_i, Y_i(\mathbf{z})\}_{i \in [n]}$. The estimated coefficients define an estimate for the function $\hat{g}$. For the variation which uses the number of treated neighbors as the covariates, setting $X_i = U_i$, the estimate is given by

$$\widehat{\text{TTE}}_{\text{LS-Num}} = \frac{1}{n} \sum_{i=1}^{n} (\hat{g}(1, |\mathcal{N}_i| - 1) - \hat{g}(0, 0)). \tag{4.5}$$

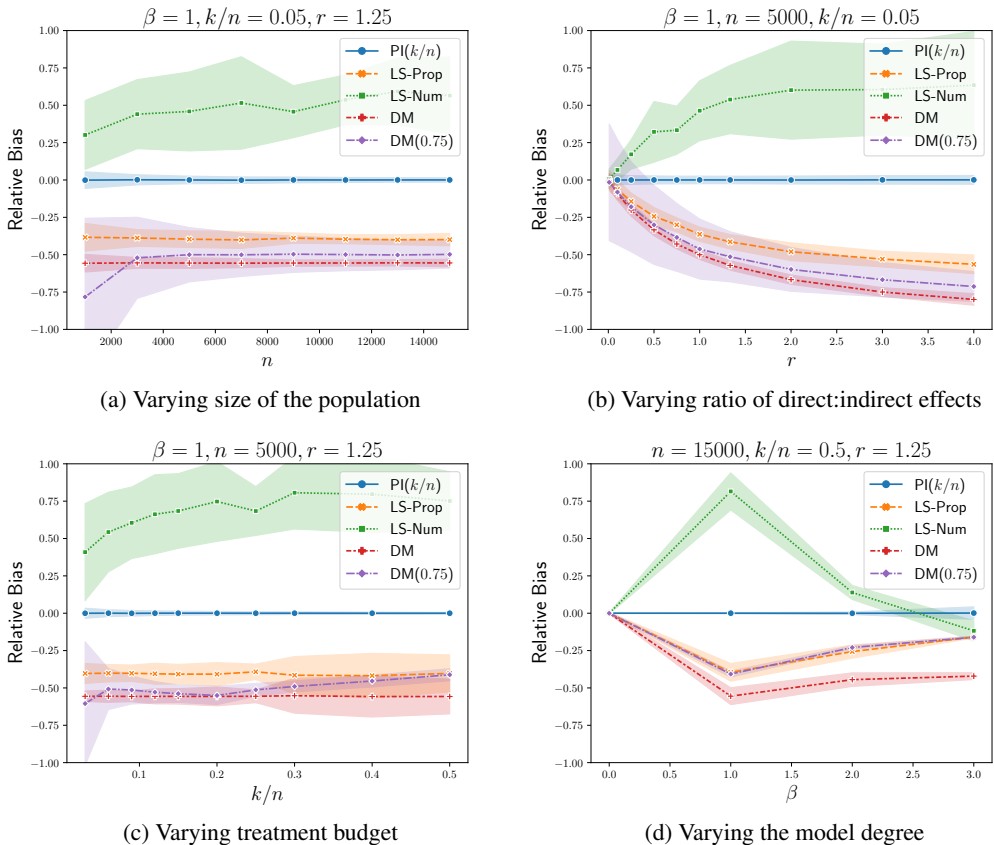

(a) Varying size of the population

(b) Varying ratio of direct:indirect effects

(c) Varying treatment budget

(d) Varying the model degree

Figure 1: Four graphs visualizing the performance of various TTE estimators as various parameters are adjusted. The height of each graph depicts the experimental relative bias of the estimator and the shaded width depicts the experimental standard deviation.

For the variation which uses the proportion of treated neighbors as the covariates, setting $X_i = U_i/(|\mathcal{N}_i| - 1)$, the estimate is given by

$$\widehat{\text{TTE}}_{\text{LS-Prop}} = \frac{1}{n} \sum_{i=1}^{n} (\hat{g}(1,1) - \hat{g}(0,0)). \tag{4.6}$$

As completely randomized design is more balanced than Bernoulli randomized design, we evaluate all the benchmark algorithms under a completely randomized design.

**Results and Discussion.** For each population size $n$, we sample $G$ networks from the distribution described above. For each configuration of parameters in the experiment, we sample $N$ treatment schedules $\{\mathbf{z}^0, \ldots, \mathbf{z}^\beta\}$ from our parameterized distribution class (Bernoulli or CRD) compute the TTE using each estimator. For each estimator, we plot the relative bias of the TTE estimates averaged over the results from these $GN$ samples and normalized by the magnitude of the TTE. The width of the shading in the figures depicts the standard deviation across the $GN$ estimates. We ran all experiments on a Linux-based machine with 20 CPU(s) and 10 cores. The experiments for the linear setting took 8.3 minutes and the experiments with varying polynomial degree took 4.6 minutes.

In Figure 1, we visualize the effect of four network/estimator parameters on the quality of each of the five TTE estimators (the four described above, and our CRD estimator with treatment targets $k_t = \frac{tk}{\beta}$). Specifically, we consider the effects of the population size ($n$), the maximum proportion of treated individuals ($k/n$), the degree of the potential outcomes model ($\beta$), and the ratio between the network and direct effects ($r$). Each of the plots fixes three of these parameters and varies the fourth. Specific settings of the parameters are listed on each plot.

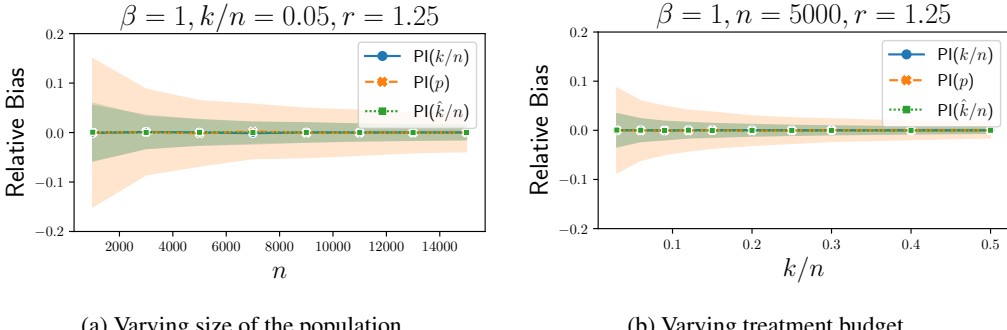

(a) Varying size of the population          (b) Varying treatment budget

Figure 2: Two graphs visualizing the performance of our proposed TTE estimators as the size of the population ($n$) or treatment budget ($k/n$) is varied. The height of each graph depicts the experimental relative bias of the estimator and the shaded width depicts the experimental standard deviation. The blue and the green plots essentially overlap.

In Figure 1, our estimator (in blue) is unbiased as expected and the variance decreases as $n$ and $k/n$ increases. However, the other estimators remain significantly biased, with higher variances than ours, regardless of treatment budget or population size. As the ratio $r$ increases the network effects become more significant relative to the direct effect, and thus the bias of other estimators also increases. As a sanity check, when the ratio is close to 0, all estimators are unbiased as there are no network effects.

In Figure 2, we compare the variants of our estimator, evaluating $\widehat{\text{TTE}}_{\text{PI}}(\mathbf{k}/n)$ under CRD and evaluating $\widehat{\text{TTE}}_{\text{PI}}(\mathbf{p})$ and $\widehat{\text{TTE}}_{\text{PI}}(\hat{\mathbf{k}}/n)$ under Bernoulli($\mathbf{p}$) randomized design, where $p_t = tp/\beta$ and $\hat{\mathbf{k}}$ is the vector of realized treatment counts. The estimators $\widehat{\text{TTE}}_{\text{PI}}(\mathbf{k}/n)$ and $\widehat{\text{TTE}}_{\text{PI}}(\hat{\mathbf{k}}/n)$ perform nearly identically. $\widehat{\text{TTE}}_{\text{PI}}(\hat{\mathbf{k}}/n)$ has lower variance than $\widehat{\text{TTE}}_{\text{PI}}(\mathbf{p})$, which is intuitive as it performs polynomial interpolation on the realized treatment fraction rather than the expected treatment fraction. We include additional experiments for higher degree models in Appendix **??**.

An additional point of comparison for these estimators is their computational complexity. Here, the most natural comparison is between our estimators and least squares, as these are the only approaches that make use of the various rounds of outcome measurements. Since our estimators require only an aggregated measurement of the individual's outcomes, the $O(\beta^2)$ runtime of the interpolation is asymptotically dominated by the $O(n\beta)$ time to read in the outcome measurements. The least squares methods fit $O(\beta)$ parameters and have time complexity $O(\beta^2 n)$.

## 5   Conclusion

We propose a new approach for causal inference under network interference which performs significantly better than existing approaches without requiring knowledge of the graph. In particular, the additional measurements from a staggered rollout design enable us to reduce the task of estimating total treatment effect to that of polynomial interpolation. We show that under a flexible class of low degree polynomial potential outcomes our estimator is unbiased with variance scaling as $O(1/n)$. Future directions include how to optimally perform model selection when $\beta$ is unknown, and generalizing to a dynamic setting by incorporating time-dependent noise to the model, considering time-varying effects, or allowing for time-varying networks. The staggered rollout design framework has implications towards estimation under other model classes beyond polynomial, such as sublinear or monotone functions, under which one may be able to construct bounds on TTE.

### Acknowledgments and Disclosure of Funding

We gratefully acknowledge financial support from the National Science Foundation grants CCF-1948256 and CNS-1955997 and the National Science Foundation Graduate Research Fellowship grant DGE-1650441. Dr. Yu is also supported by an Intel Rising Stars award and a JPMorgan Faculty Research award. Mayleen is also supported in part by the Sloan Foundation grant 90855.

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
