# OpenReview forum: "Staggered Rollout Designs Enable Causal Inference Under Interference Without Network Knowledge"
_NeurIPS.cc/2022/Conference — NeurIPS 2022 Accept_

### Official Review · Reviewer_FvE7 · 2022-06-30

**Rating:** 7
**Confidence:** 4
**Soundness:** 4 excellent
**Presentation:** 4 excellent
**Contribution:** 3 good

**Summary:**

The paper studies the problem of estimating the total treatment effect under network interference. The authors propose a method that utilizes a stagger rollout design without knowledge of the interference graph. The estimators are constructed using the Lagrange interpolation formula. The authors establish that when the potential outcomes are low degree polynomials, the proposed estimators are (nearly) unbiased with variance goes to zero. They demonstrate the performance of the estimators through simulations on synthetic data.

**Questions:**

-	Related to above, I don’t think the assumption of a low dimensional polynomial is very well motivated. The choice of low dimensional polynomial seems a bit arbitrary to me. I agree that higher degree polynomials can fully capture the structure of potential outcomes (lines 137-138), but the theoretical results can only hold for fixed beta not growing with n. Even for a beta scale like log(n), Delta_p will be small and the variance term in the theorems won’t go to zero anymore. My question is: is there any application in practice such that the low dimensional polynomial (beyond linear model) is a reasonable model for potential outcomes? Is there any way to verify this assumption in practice?
-	As pointed out by line 269-276, when the treatment probability is small, the value of interpolant at 1 can be very unstable. Is it possible to consider some other basis? For example, fourier basis, cubic splines etc.
-	How would the method perform if the low dimensional polynomial assumption is violated? It will be very interesting to see some discussions or experiments on the performance of the method under model misspecification.
-	Some typos: the equation between line 135-136: the variables should be z_j instead of x_i; line 139: “significantly smaller THAN …”


**Limitations:**

The authors have adequately addressed the limitations and potential negative societal impact of their work.

**Strengths And Weaknesses:**

Strengths

-	The paper is very well-written. The proposed method is explained in a very nice way. The theorems are stated clearly.

-	The paper addresses an interesting and important question: estimating the total treatment effect under network interference.

-	The proposed method is very elegant and simple to implement.

-	The proposed method appears to have good performance both theoretically and empirically.

Weaknesses

-	I don’t think the assumption of a low dimensional polynomial is very well motivated. Is the assumption practical in real life applications? Can we verify them in practice? (See more details in the questions part)

-	There is no discussion/experiment on the performance of the method under model misspecification.

---

> ### Author Response · Authors · 2022-08-01
> **Response to Reviewer FvE7**
>
> ## Low polynomial degree assumption
> In limited settings, it might be verifiable retrospectively for treatments that are eventually rolled out to the population at large, as one could then collect data across the entire range of treatment levels from $p \in [0,1]$. Otherwise, this assumption is actually fairly natural to motivate. As the potential outcomes model is a function of binary variables, the low degree polynomial structure is equivalent to imposing that the order of interactions in the network effects are limited to neighbor sets of size at most $\beta$. As mentioned, heterogeneous linear models are a subclass that we consider, and strictly generalize the parametric linear models used widely in empirical studies. However, linear models imply that the network effect is strictly additive across individual neighbors. A 2-degree model could naturally arise in a setting in which the network effect of two of your neighbors being treated on you depends on whether those two neighbors are also mutually friends or not. This is natural in an opinion dynamics setting in which one may discount information from friends within the same sub-community as the information may be redundant due to the mutual connection. The polynomial degree would still be expected to be low if an individual's local neighborhood decomposes into subcommunities that don't interact, such that the network effects would be additive across subcommunities. As another example, suppose a social platform is testing a “hangout room” feature that provides groups of 2 to 5 people a new environment to engage on the platform. One can posit a 5-degree model, as the change in my usage on the platform can be attributed to experience with the new feature, which in turn takes place in groups of up to 5 users.
>
> Another perspective is to observe that the low polynomial structure is primarily used to show that the expected total outcomes function is a low degree polynomial of the treatment fraction (see function $F_B$ in section 3). As the function $F_B$ represents an expectation taken over the population, where treatments are assigned uniformly at random, it is reasonable that this function varies in a smooth and simple way when the treatment level is changed. While the polynomial class is not the only hierarchy of function classes to capture complexity, it is a fairly natural one. However, we also agree that continued study of this simple approach of interpolation for staggered rollout designs for other function classes beyond polynomial would also be incredibly interesting and relevant. We hope that our work sparks research in studying causal inference amongst other similarly broad function classes as well.
>
> ## Is it possible to consider other bases?
> Yes. Our averaging of the outcomes in each time step effectively recasts the estimation problem into a curve fitting problem (as described in lines 190-194). One can use any number of approaches to tackle this. We chose polynomial interpolation as it allows for the straightforward construction of an unbiased estimator; however the other methods that you suggest can be used as well. Regarding spline interpolation, in the case where the final treatment budget is small, it seems that we are again in a setting with model misspecification, as the only contributor to the estimation of $F_D(1)$ will be the low-degree spline computed on the last few measurements.
>
> ## Model misspecification
> We agree that model misspecification is an interesting consideration that falls beyond the scope of this work. We also agree that the use of polynomial interpolation, which can be very sensitive to small changes in the polynomial degree, does not seem to be amenable to misspecification. However, this problem is not unique to our estimator or even our setting. Hyperparameter tuning is an issue that plagues many learning domains. The study of the sensitivity of these rollout estimators in this polynomial interference setting is a captivating direction for future work. In heuristic settings, we recommend choosing a conservative $\beta$ erring on lower values, as is also the typical case when using polynomial regression in supervised learning settings.

---

> > ### Comment · Reviewer_FvE7 · 2022-08-07
> > **Thank you for your response**
> >
> > I would like to thank the authors for answering my questions. I would like to keep my score after reading the authors' response and other reviewers' evaluations. Thanks!

---

### Official Review · Reviewer_PFeJ · 2022-07-11

**Rating:** 6
**Confidence:** 3
**Soundness:** 3 good
**Presentation:** 2 fair
**Contribution:** 3 good

**Summary:**

The paper proposes an unbiased and consistent estimator for the total treatment effect under network interference (under assumptions on the neighborhood that each unit's POs get affected by). The TTE can be written as a polynomial and the estimator leverages staggered rollouts to learn the coefficients of this polynomial.

**Questions:**

How does the variance of the estimator depend on p_{\beta}? In other words, how close does the final staggered rollout need to be to 1 in order for the estimator to perform well? The bound in Thm. 2 is agnostic to the actual values. I think it would useful to include p_\beta in the bound to see how it scales with that value.

Why is monotonicity of the treatments in the rollout required? A discussion of this assumption would be useful given that it leads to correlations of the outcomes across time steps. It is not clear what role it plays in the estimator or why it is needed.

**Limitations:**

The estimator only uses the outcomes of the units. However, in most applications (including the ones stated in the introduction), one would have access to covariate information which can potentially be used to improve precision. The current estimator ignores this and it is not clear that it can easily be extended to account for this.

**Strengths And Weaknesses:**

Originality and significance:
One of the main differences from prior work seems to be that the network structure for interference is not necessary, and an assumption on the neighborhood size (captured by \beta) is needed. I think the work should be interesting to the community as the staggered rollout scheme seems applicable to a variety of problems.

Clarity and quality:
Overall, the paper is well-written and the main ideas are explained clearly. There are a several typos and some notation is incomplete: so proofreading would help. For example, the PO equation after Line 135 is wrong (it should be in terms of z, j, and j'). In Line 123, d_{out} is not defined.

---

> ### Author Response · Authors · 2022-08-01
> **Response to Reviewer PFeJ**
>
> ## Variance of the estimator w.r.t. $p_{\beta}$
> Thanks for the question, and we're happy to add a discussion for this into the paper. The limiting factor in the bounds with respect to the dependence on the treatment schedule is the quantity $\Delta_{\bf p}$, which is the minimum interval length between consecutive treatment levels. However, given a budget $p_{\beta}$ for the maximum treatment level, the treatment schedule that would optimize the bounds would be to choose a consistent treatment schedule with $p_t = t p_1$, such that $\Delta_{\bf p} = p_\beta / \beta$. So one could substitute $p_\beta / \beta$ in lieu of $\Delta_{\bf p}$ in the bound for Thm 2.
> Our experimental results do illustrate the variance decay for our estimator as a function of the treatment budget (see Figure 1(c) for completely randomized design and Figure 5(c) in Appendix D for Bernoulli design).
>
> ## Monotonicity assumption
> The monotonicity assumption is not required (mathematically) to use the estimator. Rather, it is a constraint that is present in many real-world scenarios where such an estimator would be useful. Treatments in medication trials can have life-altering, irreversible effects (for better or worse), and the exposure of individuals to an advertising campaign cannot be “taken back” or reversed. In these settings, the experimental designer only has the option to introduce treatment to new individuals, imposing this treatment monotonicity. In other domains, such as the rollout of new interfaces on social media platforms, treatments are reversible and it makes sense to think about a treatment schedule unconstrained by monotonicity. This would likely make the calculations easier, as one could then sample treatment vectors that are actually independent across time, whereas under monotonicity, we have to additionally account for the dependence in the treatment variables across time in the calculations.

---

> > ### Comment · Reviewer_PFeJ · 2022-08-07
> > **Thanks for your response**
> >
> > I thank the authors for answering my questions. My evaluation of the paper remains the same.

---

### Official Review · Reviewer_JUrX · 2022-07-11

**Rating:** 7
**Confidence:** 4
**Soundness:** 3 good
**Presentation:** 3 good
**Contribution:** 3 good

**Summary:**

The paper proposes a staggered rollout randomized design for estimating the total treatment effect with network data. The proposed estimator does not require knowledge of the network. Based on an assumption of neighbourhood  interference (low degree interactions among neighbours of a node), the total treatment effect (TTE) is estimated unbiasedly along with its variance. The key theoretical result depends on a relation between staggered rollout and polynomial extrapolation. The results on synthetic data show superior performance of the proposed method against baselines.

**Questions:**

Following are my queries for the author(s)

(A) Any heuristics why do the count k_1 only appears in the bound in Theorem 3 for completely randomized design with distinct treatment counts?

(B) Is there a scope of considering sparse networks (which would essentially put some restriction on the degree of the network) as real world networks are often sparse?

(C) With staggered rollout, How do the author(s) take into account the time varying effect that may gradually come in the network with time?

**Limitations:**

Yes the authors adequately addressed the limitations and potential negative societal impact of their work.

**Strengths And Weaknesses:**

strengths

(A) staggered rollout random design to estimate TTE
(B) Novel use of Lagrange interpolation along without staggered design to propose the estimator of TTE
(C) The knowledge of the network structure is not necessary for the TTE estimator
(D) Key theoretical results showing the unbiasedness of the proposed estimator along with $O(1/n)$ variance

weaknesses


(A) The necessity of low polynomial degree assumption (Assumption 2) is not entirely clear to me. A bit more discussion about the requirement of this assumption would have been useful.
(B) Some time complexity results of the proposed potential outcomes model against the least square regression would be great!
(C) A separate subsection in Section 3 with the theorems, results and discussions on them would be appropriate

---

> ### Author Response · Authors · 2022-08-01
> **Response to Reviewer JUrX**
>
> ## Necessity of Assumption 2 (low polynomial degree)
> The low polynomial degree property can be viewed in two perspectives. As we have it defined, the polynomial structure is with respect to the potential outcomes function, which is over a binary vector of variables (rather than a function over real-valued scalars). Thus, low degree polynomial structure is perhaps more accurately conceptualized as a constraint on the order of interactions amongst neighbors of an individual. As such, the polynomial degree $\beta$ constrains that the order of interactions amongst the neighbors is at most $\beta$. An alternate perspective is to see that the low degree polynomial structure on the potential outcomes function also implies that the expected total population outcomes as function of the overall treatment level is a low degree polynomial (see function $F_B$ in section 3). This is perhaps easier to conceptualize and also gives insight why "low degree" enables more efficient estimation. If the polynomial degree of this function were high, this would mean that the overall population outcomes vary in a complex way when we slightly change the treatment fraction.
>
> ## Time complexity results
> Thank you for this suggestion. Since our estimators require only an aggregated (averaged) measurement of the individual’s outcomes, the $O(\beta^2)$ runtime of the interpolation is asymptotically dominated by the $O(n\beta)$ time to read in the outcome measurements. The least squares methods that we compare against fits $O(\beta)$ parameters and has time complexity $O(\beta^{2n})$.
>
> ## Seperate subsection in section 3
> We agree that adding a subsection header at line 222 will help to distinguish our main results; we will put this edit into the final version.
>
> ## $k_1$ in Bound in Theorem 3
> This is mainly a result of the computation in Lemma 9. Initially, since the covariance terms describe relationships between two time steps, they are more naturally expressed as functions of $\Delta$. This lemma uses asymptotics to bound each covariance term by a function of its earlier time step. Monotonicity allows us to further bound all of these by the expression for timestep $1$, explaining the $k_1$.
>
> ## Sparse networks
> Sparse networks implicitly constrain the complexity of the model in our setting, since $\beta \leq d_{\text{max}}$, so they are actually beneficial to estimation as they simplify the model. Accordingly, our results can generally handle sparse networks. In particular, families of networks with degree $o(\log n)$, a reasonable assumption in social networks, have asymptotically decaying variance for our estimators.
>
> ## Time varying effect in networks
> In this work, we limit our attention to static network effects. Careful consideration of how to parameterize time varying effects within our polynomial framework is needed to extend our results in this direction. This is certainly a desirable avenue of future study. When the total treatment budget $p$ is small, such that a significant number of individuals are observed under the baseline outcomes across all stages of the experiment, then a simple modification could handle time-fixed effects: use these baseline individuals to estimate the time-fixed effects and then subtract them from the current estimator. This would not fundamentally change the estimator or results in any significant way, and we would be happy to include this extension as well.

---

### Official Review · Reviewer_nkMf · 2022-07-16

**Rating:** 4
**Confidence:** 4
**Soundness:** 4 excellent
**Presentation:** 3 good
**Contribution:** 2 fair

**Summary:**

The text explores estimation under a rollout set-up, whereby the treatment of units is sequentially increased over time. This is a common setting in tech companies for example, also known as a “ramp-up” period. By assuming a polynomial function of interference (in the treatment status of the neighbors), they suggest lagrange interpolation as an estimator, which, nicely, does not require knowledge of the graph.



**Questions:**

N/A

**Limitations:**

See above.

**Strengths And Weaknesses:**

The paper is nicely written and easy to follow. There are some synthetic experiments that illustrate their point, as well as a few technical results that follow nicely from their setting. The biggest strengths are

1. The usefulness of a graph agnostic estimator. Knowing graphs exactly is rare and proposing a graph agnostic estimator is useful in these settings (and uncommon in the literature on interference)
2. The idea of leveraging roll outs is understudied for how valuable they can be to study interference.

My biggest criticisms are:
1. The related works section is short on details. Namely, I’m not sure I completely grasp the improvement over [25]. Going from linear to polynomial does not seem to be a hugely difficult step, so it would help if the authors could clarify in what way this extension is non-trivial.

2. The roll-out model assumes no time-dependent noise (or noise of any kind). This is, I imagine, what leads to the (surely true but strange) result of Theorem 6. That more observations does not reduce variance is somewhat of a red flag that the model & estimator would not be so useful in practice. Adding noise terms to these models would make for a stronger more practical estimator.

3. Finally, and this is less crucial, knowing beta is unlikely to hold in practice. And even so, ramp-ups/roll-outs rarely have more than 5-6 stages, so it is unlikely we would ever be able to estimate a polynomial of large degree, which would be necessary if neighborhoods are large. Of course, low degree polynomials can approximate large degree ones, but I did not see any results about the quality of these approximations. It is also my impression that Lagrange interpolation is not very robust to these kinds of mispecifications.

---

> ### Author Response · Authors · 2022-08-01
> **Response to Reviewer nkMf**
>
> ## Improvement over [25]
>
> The polynomial structure is with respect to the potential outcomes function, which is over a binary vector of variables. Thus, low degree polynomial structure is perhaps more accurately conceptualized as a constraint on the *order* of interactions amongst neighbors of an individual. As such, the linear model is particularly simple as it can equivalently be defined as a model satisfying additivity of network effects. The estimator in the linear setting can be rearranged to show that it is simply the sum over independent random variables, which is certainly easy to analyze. The extension from $\beta=1$ to $\beta > 1$ goes from having no interactions (purely additive over treated neighbors), to having non-trivial interactions amongst those in the neighborhood set. This fundamentally requires a new analysis in order to show unbiasedness and bounds on the variance. Additionally, a key contribution is showing that multiple measurements over time can reduce a complex network causal inference problem into a simple interpolation task (in our case polynomial, though the meta-idea is not limited to the polynomial function class). This insight would not have been clear in the linear setting as the setup in [25] is stated simply as a single shot experiment setting with historical/prior estimates on the baselines.
>
> ## Adding noise to the model
> We agree that considering noise is instrumental in any real world application of this style of estimator. However, due to the space constraints, we excluded this because we felt that it was a distraction from the main narrative: the connection between rollout designs and polynomial extrapolations. Incorporating mean-$0$ time-independent noise is easy to do and would not significantly change the results, especially because the estimator is a linear function of the measurements. The current estimator would still be unbiased, and the variance calculations would be slightly modified to add a lower order term which arises from standard calculations. We would be happy to include this in the camera-ready version. Incorporating time-dependent noise would require more careful modeling considerations, but is a great suggestion for future research. When the total treatment budget $p$ is small, such that a significant number of individuals are observed under the baseline outcomes across all stages of the experiment, then a simple modification could handle time-fixed effects: use these baseline individuals to estimate the time-fixed effects and then subtract them from the current estimator. This would not fundamentally change the estimator or results in any significant way, and we would be happy to include this extension as well.
>
> ## Model misspecification
> We agree that model misspecification is an interesting consideration that falls beyond the scope of this work. We also agree that the use of polynomial interpolation, which can be very sensitive to small changes in the polynomial degree, does not seem to be amenable to misspecification. However, this problem is not unique to our estimator or even our setting. Hyperparameter tuning is an issue that plagues many learning domains. The study of the sensitivity of these rollout estimators in this polynomial interference setting is a captivating direction for future work. In heuristic settings we recommend choosing a conservative $\beta$ erring on lower values, as is also the typical case when using polynomial regression in supervised learning settings.

---

### Meta-Review · Area_Chair_2F5z · 2022-08-27

**Recommendation:** Accept
**Confidence:** Certain

**Metareview:**

The paper proposes a staggered rollout randomized design for estimating the total treatment effect with network data, without requiring the knowledge of the network. This is an interesting problem that appears in many real world settings. The paper has many strengths and few weaknesses. I believe that the weaknesses can be addressed based on the authors' response. I strongly advise the authors to revise the manuscript based on the received comments when submitting the final version.

**Award:**

No

---

### Decision · Program_Chairs · 2022-09-14

Accept